

# Dissection of genetic architecture for tiller angle in rice (*Oryza sativa*. L) by multiple genome-wide association analyses

Muhammad Abdul Rehman Rashid[1,2,3], Rana Muhammad Atif[4,5], Yan Zhao[1,6], Farrukh Azeem[3], Hafiz Ghulam Muhu-Din Ahmed[7], Yinghua Pan[8], Danting Li[8], Yong Zhao[1], Zhanying Zhang[1], Hongliang Zhang[1], Jinjie Li[1] and Zichao Li[1]

[1] State Key Laboratory of Agrobiotechnology/Beijing Key Laboratory of Crop Genetic Improvement, College of Agronomy and Biotechnology, China Agricultural University, Beijing, China
[2] School of Agriculture, Yunnan University, Kunming, Yunnan, China
[3] Department of Bioinformatics and Biotechnology, Government College University Faisalabad, Faisalabad, Pakistan
[4] Department of Plant Breeding and Genetics, University of Agriculture Faisalabad, Faisalabad, Pakistan
[5] Precision Agriculture and Analytics Lab (PAAL), National Center for Big data and Cloud computing (NCBC), University of Agriculture Faisalabad, Faisalabad, Pakistan
[6] College of Agronomy, Shandong Agricultural University, Taian, Shandong, China
[7] Department of Plant Breeding and Genetics, Islamia University, Bahawalpur, Bahawalpur, Pakistan
[8] Rice Research Institute, Guangxi Academy of Agricultural Sciences, Guangxi, China

Corresponding author
Zichao Li, lizichao@cau.edu.cn

## ABSTRACT

**Background:** The rice plant architecture is determined by spatially and temporally domesticated tiller angle. The deeper insight into the genetic mechanism for rice plant architecture will allow more efficient light capture by increasing the planting density, reproducibility, and the ability to survive in a stressful environment.
**Methods:** In this study, a natural population of 795 genotypes further divided into *japonica* and *indica* subpopulations, was evaluated for tiller angle. A significant variation with a wide range was observed. Genome-wide association analysis was performed by the general linear model (GLM), and compressed mix linear model (cMLM) for three populations to disclose the genomic associations. The population principal components and kinship matrix in 1,000 permutations were used to remove the false positives. The candidate genes were evaluated for their functional annotations and specific molecular pathways. The sequencing-based haplotype analysis was further performed to reveal the functional variation among candidate genomic regions.
**Results:** As a result, 37 significant QTLs with 93 annotated loci were identified. Among the loci, a known tiller angle controlling locus *TAC1* was also identified. The introduction of the sequence pooling technique was observed fruitful to screen the 12 significant QTLs with 22 annotated loci. For ten of these loci, the functional variations were identified by haplotype analysis. These results were not only providing a better understanding of the genetic bases of rice plant architecture but also provide significant information for future breeding programs.

## INTRODUCTION

Among the cereal crops, rice (*Oryza sativa* L.) is one of the most important food staples. Its role as a model plant to study the monocots is due to its comparatively collinear and smaller genome (*Li et al., 2003*). To feed the continuously growing human population, the increasing crop yield is always an objective of any research activity. The main culm and tillers play a seminal role to determine the rice plant architecture and finally the yield (*Wang & Li, 2005*). Plant architecture referred to the three-dimensional arrangement of arial parts of plants, is a major factor contributing to the total plant yield (*Reinhardt & Kuhlemeier, 2002*; *Wang & Li, 2006*). The plant architecture can be determined by the variation in tillering, plant height, leaf shape and arrangement and panicle morphology (*Wang & Li, 2005*). Tiller angle affects the ability of survival and to trap maximum light for photosynthesis. The erect to optimum spreading tillers of plants have been evolved during rice domestication from wild species to cultivated varieties while a relatively wider tiller angle in *indica* sub-species has been observed than in *japonica*. In dense planting, a wider tiller angle may enable the plant to escape some diseases by reducing the humidity. Contrastingly, it will cover more space and increase the leaf shading and can cause reduction in photosynthesis. The dense and upright leaves have also been reported to improve the leaf nitrogen accumulation for grain filling (*Wang & Li, 2008*).

As with other yield related traits, tiller angle also showed a complex genetic behavior. In previous studies different quantitative trait loci (QTLs) and genes have been identified for tiller angle as *Ta* (*Li et al., 1999*), *OsHOX1*, *OsHOX28* (*Hu et al., 2020*), *qTA-9a* (*Qian et al., 2001*), *qTA-9b* (*Qian et al., 2001*), *qTA9-2* (*Shen et al., 2005*) and *qTA-9* (*Yu et al., 2005*), and *qTAC-8* (*He et al., 2017*) have been mapped. The *TAC1* (*Yu et al., 2007*) have been mapped and cloned as well, while the *TAC3* together with *TAC1* and *D2* have been known for significant variation in tiller angle (*Dong et al., 2016*; *Li et al., 2021*). A study revealed the effect of overexpression of CpG-binding protein gene *OsMBD707* causing a larger tiller angle (*Qu et al., 2021*). Several mutants such as *La* (*Singh, Multani & Khush, 1996*), *La2* (*Li et al., 2003*), *er* (*Yu et al., 2007*), *D20* (*Kinoshita, Takahashi & Takeda, 1974*) and *Spk(t)* (*Miyata et al., 2005*) were also observed to affect the tiller angle. The TCP transcriptional factor encoded by *TIG1* was found to affect the tiller angle in rice (*Zhang et al., 2019*). Another QTL mapping study for tiller angle performed by RIL population identified the 24 alleles on 12 QTLs (*Zeng et al., 2019*).

Various genetic studies for plant growth regulators including the *TAC4* (*Li et al., 2021*) and *OsmiR167a* gene (*Li et al., 2020*) have also been reported to control tiller angle by regulation and endogenous distribution of auxin contents. Furthermore, *LAZY1* is known to control the tiller angle by gravitropism (*Li et al., 2007*; *Yoshihara, Spalding & Iino, 2013*; *Zhu et al., 2020*) and asymmetric distribution of Auxin (*Zhang et al., 2018*), and *LAZY2* by regulation of starch biosynthesis in gravity-sensing cells (*Huang et al., 2021*). The phytochrome-interacting factor-like protein *OsPIL15* has also been reported to

integrate the light and gravtropism to regulate the tiller angle in rice (*Xie et al., 2019*). The hormonal homeostasis in combination with environmental factors has also been studied to involve in regulation of tiller angle (*Lomax, 1997*). Recently, strigolactone hormone has been reported to regulate tiller angle by shoot gravitropism in rice through auxin biosynthesis inhibition (*Sang et al., 2014*). Several tiller angle governing loci in rice have been identified by both classical and molecular methods. Although the mapping, cloning and functional genetics studies laid a solid basis to understand the plant architecture development exact mechanisms are still unclear.

The genome-wide association (GWA) study with the high throughput sequencing technology has appeared as an efficacious tool to identify the QTL/gene associated with complex agronomic traits. The GWA-mapping took the advantage of maximum recombinant events to identify the trait association to the narrower genomic region. The genome-wide distribution of single nucleotide polymorphic (SNP) markers enhanced the power of GWA mapping studies. In recent years, the GWA study has gained the attention of rice researchers to identify different agronomically important traits including flowering time and yield traits (*Huang et al., 2010*; *Huang et al., 2012*), stress related traits as salinity (*Kumar et al., 2015*) and drought (*Al-Shugeairy, Price & Robinson, 2015*) but no specified GWA study has performed to dissect the genetic basis of tiller angle. This study was aimed to investigate the genome wide regions associated with tiller angle in rice. It will not only extend our understanding about the genetic nature of tiller angle in a model plant but also will contribute toward improving yield potential in breeding programs.

## MATERIALS AND METHODS

### Plant and field material

The 795 geographically diverse accessions from 3,000 Rice genome project (3KRGP) (*Li, Wang & Zeigler, 2014*; *Li et al., 2014*) were used in this study. The genetic homozygosity was ensured by grown plants through single seed descent for three years. All the plant material was cultivated in two blocks at a bird-net protected experimental farm of China Agricultural University, Sanya city, Hainan (18°N, 109°E) province of China. Three lines of 1 m length and seven seed per line were grown. Five guarded plants from middle line were selected for phenotyping. All the standard cultural practices were kept constant.

### Phenotyping

The tiller angle (TA) of five guarded plants from each experimental block was recorded as the angle between extreme right and extreme left tiller (*Chen et al., 2012*) by goniometer. And, an average value of ten individual plants was considered as final TA for further analysis (Table S1).

### Genotyping and population structure

The second generation of whole-genome high-throughput sequence with ~13fold coverage was obtained by Illumina sequencing and the sequence reads were aligned against rice reference genome. The direct comparison was performed with corresponding regions of *japonica* cv. Nipponbare reference sequence to produce 10 million raw SNPs.

The screening for >50 percent missing SNPs resulted in 3.3 million SNPs for 795 genotypes. These SNPs covered the 373 Mb (mega-base) of whole rice genome with an average SNP density of 12.3 SNPs per kb (kilo-base). All the selected SNP markers distributed over the 12 chromosomes were used for detection of population structure. The principal components and kinship matrix were calculated by GAPIT program (http://www.maizegenetics.net/GAPIT). On the basis of principal components and kinship matrix, whole population was divided into two sub-populations; *japonica* and *indica*. Then, the principal components and kinship matrix were re-estimated for individual sub-population to use as covariates in the mixed linear model (MLM). The physical position of known genes and candidate loci were acquired from rice online data base (http://rapdb.dna.affrc.go.jp).

## Statistical analysis for phenotypes

The analysis of variances (ANOVA) was performed to estimate the genotypic variance (Vg), error variance (Ve) and interaction variances by using the STATISTICA software (StatSoft 1995; Tulsa, OK, USA). The phenotypic data descriptive and Q-Q plots were analyzed by SPSS version 19 (http://www-01.ibm.com/software/analytics/spss/).

The broad sense heritability ($H^2$) and repeatability ($R^2$) (*Sukumaran et al., 2015*) were estimated by using genotypic variance (Vg), error variance (Ve) and interaction variances according in following formula:

$$R^2 = \frac{\sigma^2 G}{\sigma^2 G + \dfrac{\sigma^2 (G \times E)}{b} + \dfrac{\sigma^2 E}{br}}$$

where $R^2$ is the estimate of repeatability, $\sigma^2 G$ is the genetic variance, $\sigma^2 G \times E$ is the genotype to blocks interaction variance, and $\sigma^2 E$ is the residual variance, r is the number of replications, and b is the number of blocks. All the related variances were estimated by ANOVA.

The data normality was confirmed by Q-Q plot drawn by SPSS program and also in GAPIT analysis. The best linear unbiased predictions (BLUPs) were estimated in GAPIT analysis using R-program.

## Genome wide association analysis

The average tiller angle of ten plants for each genotype were used as final tiller angle. Two models (GLM, cMLM) were applied on three populations (Whole population, *indica* and *japonica* sub-populations) by GAPIT software to identify the SNPs association with tiller angle. In first model, the population structure (PCAs) was incorporated to accomplish the general linear model (GLM). In second model, population structure and kinship matrix respectively were included as fixed and random effect to conduct compressed mix linear model (cMLM). The 1,000-permutations test by MLM was performed for detection of false SNP signals.

## Sequence pooling, candidate genes identification and haplotyping

The QTLs regions were defined by SNP significance at threshold of $-\log 10\ (P) \geq 4$ which were further screened by deletion of SNPs not reside in CDS or promoter regions of annotated loci. For further identification of significant loci, a new sequence based pooling strategy was applied. The 20 genotypes with highest phenotypic value and 20 genotypes with lowest phenotypic value were selected separately from two subpopulations. The difference of major and minor allele frequencies in four pools were compared by $\chi^2$-test at $\alpha = 0.05$. The SNPs with significantly different major and minor allele frequency were selected to define significantly associated loci. Then all the significant and non-significant SNPs in promoter and CDS region of selected loci were used to construct a map and to evaluate their functional haplotypes.

# RESULTS

## Phenotypic variation of tiller angle

The highly divergent worldwide collection of 795 rice genotypes showed significant variation for tiller angle ranging from 20.1 to 82.7 degrees (Table S1). The mean tiller angle for *japonica* sub-population (38.05 ± 7.75) was significantly lower than that of *indica* sub-population (41.84 ±9.8) and whole populations (40.55 ± 9.3) (Table 1). An identical and positively semi-skewed frequency distribution was observed for all three populations (Fig. 1, Table 1). The two-way analysis of variance (ANOVA) showed the significantly high variation among genotypes (Table 2). The high broad sense heritability ($h^2$) and the repeatability factor ($R^2$) demonstrated the genotypes as the main source of variation for tiller angle which was consistent in three populations. The *japonica* cultivars showed relatively lower heritability but the higher repeatability confirmed the genotypic effect as the main cause of variation.

## Population structure and relative kinship

The evaluation of 795 genotypes against 3.3 million SNPs by PCA and kinship matrix resulted in a strong population structure. The PC1 showed the maximum variation which divided the whole population into two clearly distinct groups. The same results were observed by phylogenetic tree and kinship matrix (Fig. 2). The group-I was composed of 289 *japonica* accessions while the group-II framed the 506 *indica* genotypes and named as *japonica* and *indica* subpopulations respectively (Table S1). The other structural features were consistent as previously reported (*Zhang et al., 2009*; *Zhang et al., 2011*).

## Association mapping and QTL analysis

Firstly, the SNPs association was estimated by GLM and MLM models for *japonica*, *indica* subpopulations and whole population separately (Fig. 3). Then, a 1,000-time permutation with randomized phenotype was performed by MLM and the threshold $-\log 10\ (p) \geq 4$ was set for calling the significant associations. It is suggested that more association signals in a common region are caused by linkage disequilibrium (LD) decay while the lower LD decay rate can reduce the power of association analysis for complex traits.
The genome-wide $r^2$ reduction for *japonica* and *indica* subpopulations to 0.28 and 0.25
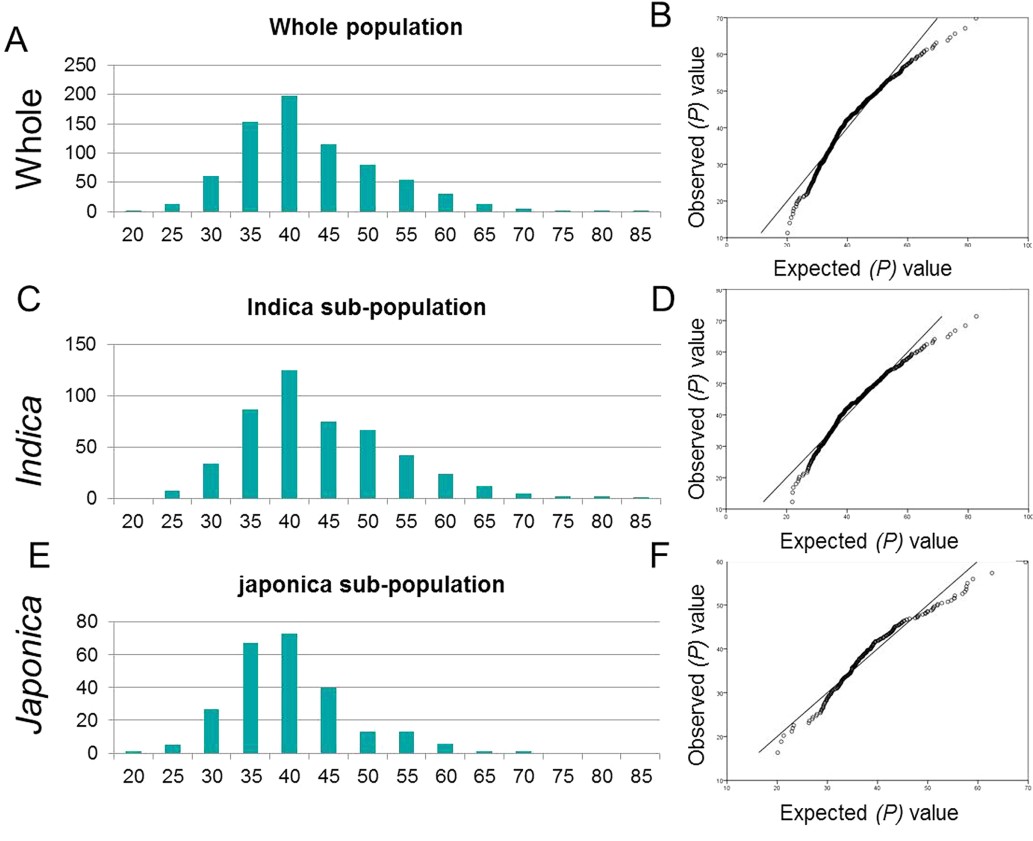

**Figure 1 Frequency distribution and Q-Q plots.** (A, B) Whole population (C, D) for *indica* sub--population and (E, F) *japonica* sub-population.

with LD decay rate 167 kb and 123 kb has been reported previously (*Huang et al., 2010*; *Huang et al., 2012*). Therefore, in this study, the genomic region containing minimum three significant SNPs within 170 kb range was defined as a QTL. In this way, the 37 QTLs were defined by 336 significant SNPs (Tables S2 & S3). Among these QTLs, 13 and nine QTLs were specifically identified in *indica* and *japonica* subpopulations while 15 non-common QTLs were observed in whole population (Fig. 4, Table 3).

Notably, *TAC1*, a previously identified gene was found as a candidate of *qTA9-6*. The *TAC1* was previously mapped for tiller angle in rice by using $F_2$ segregating population originated from a cross between *indica* genotype and *japonica* introgressed line (*Yu et al., 2007*). Its molecular study also has revealed the functional single nucleotide polymorphism (*Jiang et al., 2012*). Here in our results, two and three significant SNPs were found in promoter and in genomic region of *TAC1* respectively.

## Candidate genes evaluation

To evaluate the available candidate loci in identified QTLs, the SNPs in physical range of annotated loci or their promoter were observed. The 97 and 43 significantly associated signals were identified in promoter and coding sequence (CDS) regions of 93 annotated loci. Those may consider as candidate loci for the respective 28 QTLs.

**Table 1 Descriptive statistics of tiller angle in different populations.**

| Populations | N | Range | Mean ± SD | CV% | h² (%) | Skw | Kur | r²* |
|---|---|---|---|---|---|---|---|---|
| Whole population | 795 | 20.10–82.70 | 40.55 ± 9.33 | 23.01 | 73 | 0.92 | 1.28 | 0.91 |
| *Indica* | 514 | 21.90–82.70 | 41.84 ± 9.80 | 23.42 | 74 | 0.85 | 1.01 | 0.91 |
| *Japonica* | 281 | 20.10–69.50 | 38.05 ± 7.75** | 20.37 | 45 | 0.87 | 1.34 | 0.92 |

Notes:
Skw, Skewness.
Kur, Kurtosis.
* $r^2$ = repeatability
** Two-tail T-test significance at $P \geq 0.01$.

**Table 2 Analysis of variance in whole population for tiller angle.**

| Sources of variation | Degree of freedom | Sum squares | Mean squares | F-value |
|---|---|---|---|---|
| Replications | 4 | 1,746.26 | 436.57 | 10.56** |
| Treatment | 1,589 | 8,16,231.99 | 513.68 | 12.43** |
| Genotypes | 794 | 6,93,201.20 | 873.05 | 21.12** |
| Block | 1 | 43.19 | 43.19 | 1.05[ns] |
| B × G | 794 | 1,22,987.61 | 154.90 | 3.75** |
| Error | 6,360 | 2,62,935.34 | 41.34 | |
| Total | 7,949 | 1,080,913.60 | | |

Note:
** Significance at $\alpha = 0.01$; ns, non-significant; b, blocks; g, genotypes.

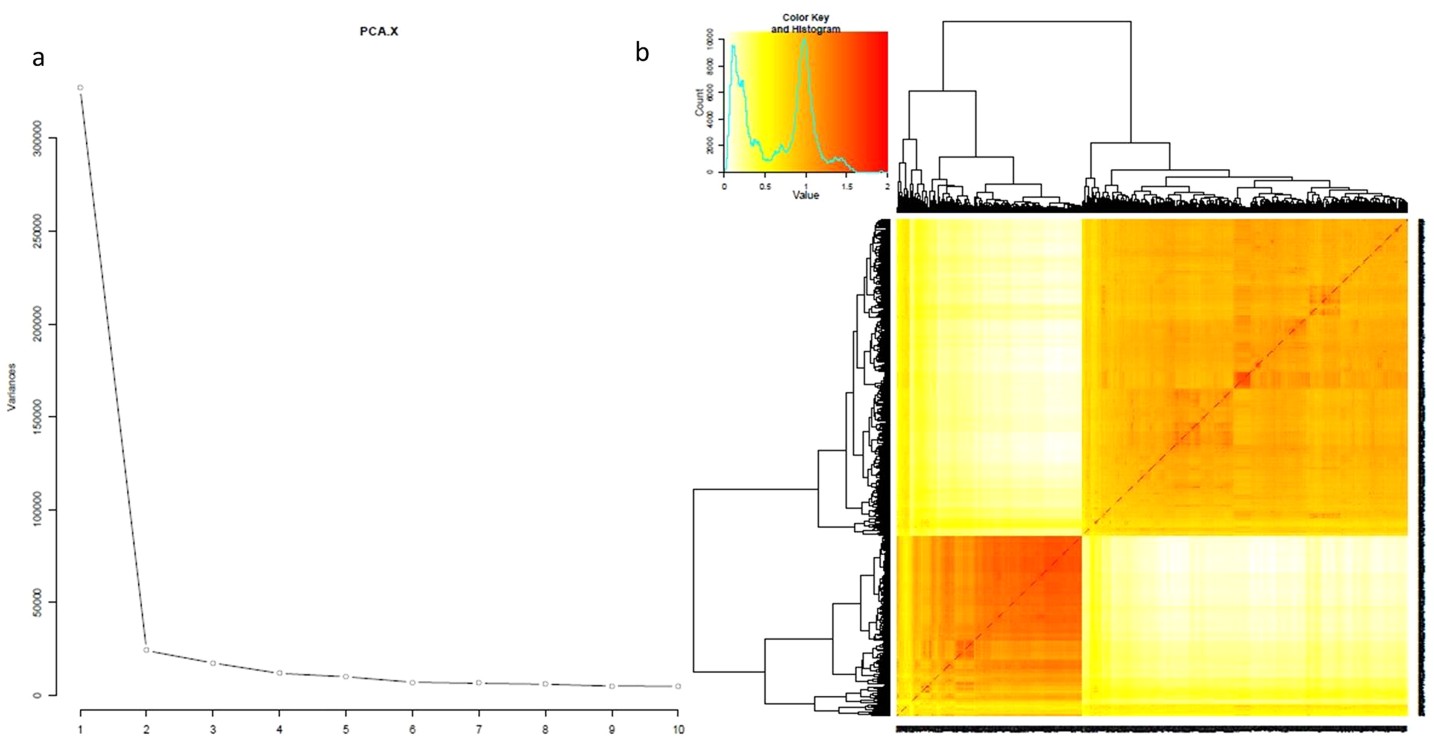

**Figure 2 Population structure analysis for 795 genotypes.** (A) PCAs, (B) kinship.               

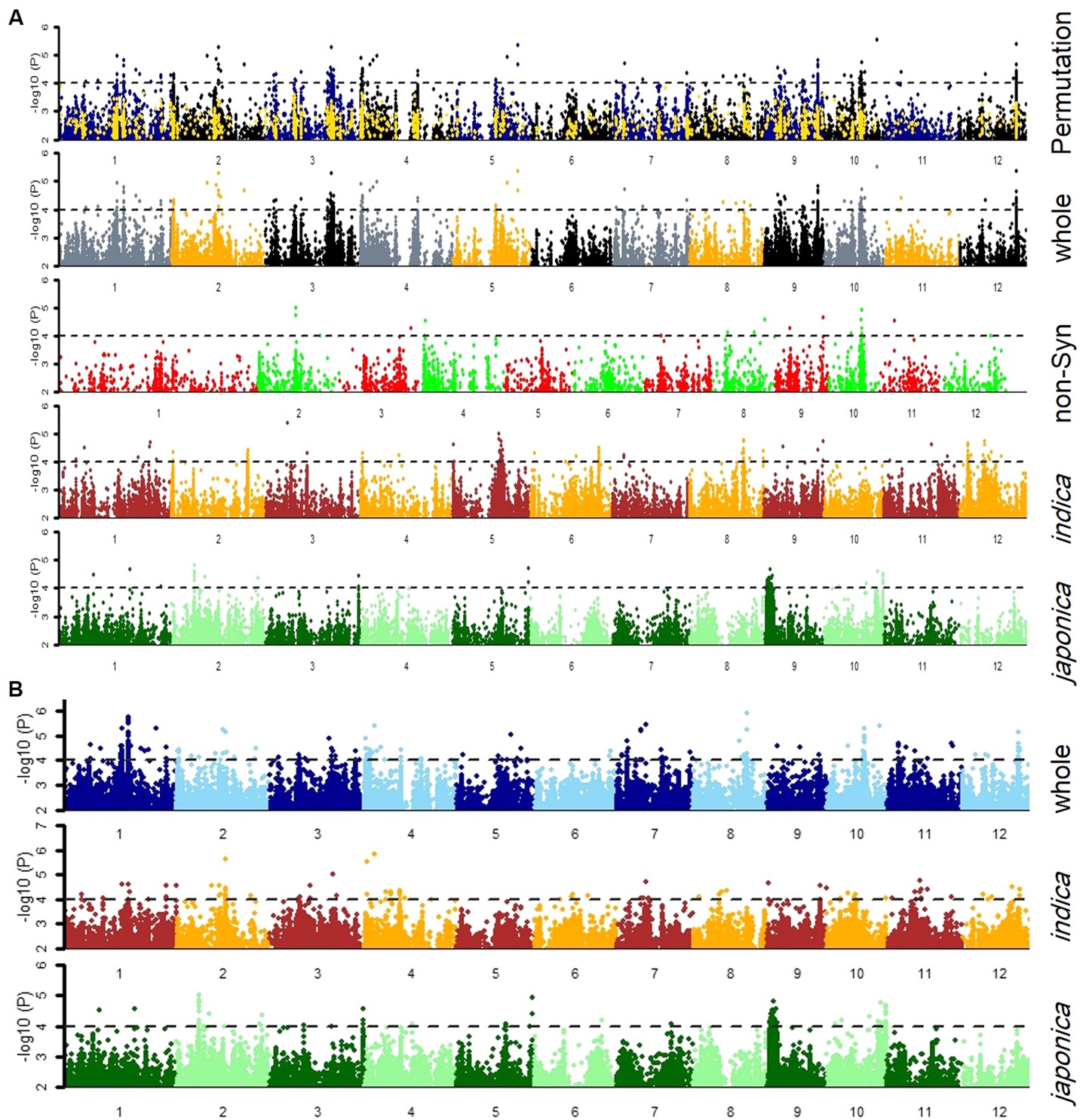

**Figure 3 Manhattan plots for tiller angle for SNPs in the whole population, and *indica* and *japonica* subpopulations SNPs obtained by (A) cMLM and (B) GLM, and Non-Synonymous SNPs by (A) cMLM.** The blue and black dots in the permutation plot indicate the original *p*-value of SNPs while gold dots indicate the 1,000-permutation *p*-value.

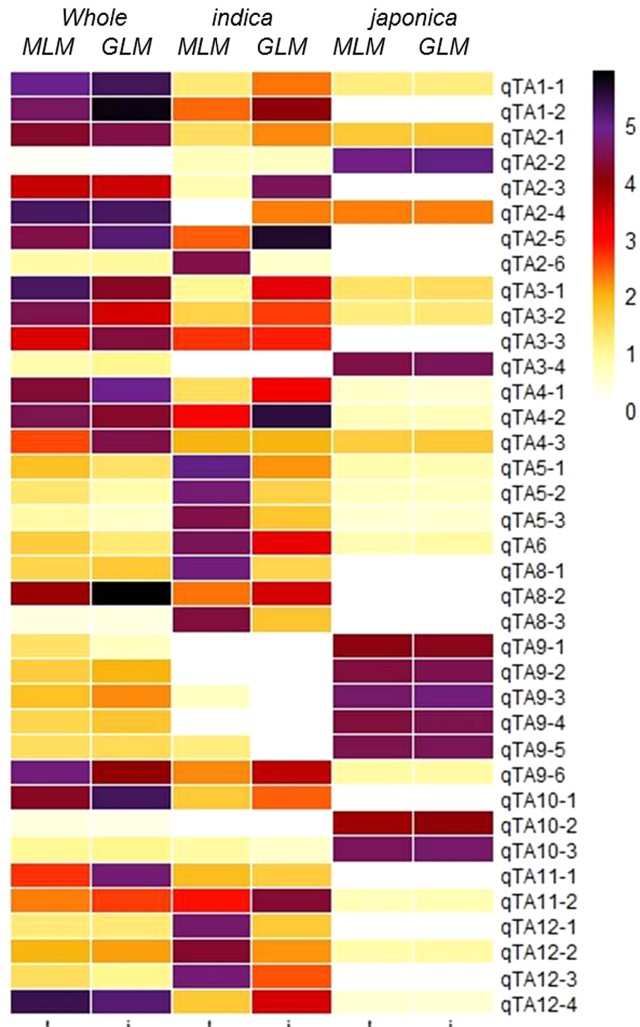

**Figure 4 Heatmap of 37 GWA-QTLs for –log10(P) of peak SNPs in different association models.**

We also introduced and adopted the sequence pooling approach for screening the significant genes. For this purpose, high phenotype pool and low phenotype pools were constructed by Selection of 20 genotypes from right and left tails of frequency distribution lines of *indica* and *japonica* subpopulations separately (Table S4). The major and minor allele frequencies of high and low pools were compared and significance was estimated according to $\chi^2$ (chi-square) test. Finally, we sieved the 28 SNPs with significant difference in major and minor alleles to reveal the 22 candidate loci for tiller angle (Table S5). Among them, 13 and seven loci were specific for *indica* and *japonica* subpopulations while two loci commonly presented by both subpopulations.

## Haplotypes evaluation for significant loci

We further evaluated the 22 candidate loci in 12 QTLs for functional haplotypes. The haplotypes evaluation by using CDS and promoter residing SNPs was conducted and

**Table 3 The list of QTLs identified in different models of association mapping for tiller angle.**

| GWAS-QTL | Chromosome | Model-population | Peak SNP | Flanking SNPs | |
|---|---|---|---|---|---|
| | | | | Right | Left |
| qTA1-1 | 1 | MLM/GLM-whole | 22,379,048 | 21,952,887 | 22,436,318 |
| qTA1-2 | 1 | MLM/GLM-whole | 24,887,286 | 24,781,381 | 25,029,148 |
| qTA2-1 | 2 | MLM/GLM-whole | 8,29,685 | 8,29,685 | 8,70,392 |
| qTA2-2 | 2 | MLM/GLM-Jap | 8,709,646 | 8,686,587 | 8,709,646 |
| qTA2-3 | 2 | GLM-ind | 16,631,825 | 16,631,825 | 16,766,562 |
| qTA2-4 | 2 | MLM-whole | 18,056,292 | 17,094,194 | 18,789,028 |
| qTA2-5 | 2 | GLM-ind | 18,789,028 | 18,702,054 | 18,789,028 |
| qTA2-6 | 2 | MLM-ind | 29,311,370 | 29,221,199 | 29,407,400 |
| qTA3-1 | 3 | MLM-whole | 25,422,986 | 25,263,068 | 25,423,816 |
| qTA3-2 | 3 | MLM-whole | 26,277,666 | 26,263,239 | 26,313,715 |
| qTA3-3 | 3 | GLM-whole | 33,199,574 | 33,198,645 | 33,199,696 |
| qTA3-4 | 3 | GLM-jap | 35,863,137 | 35,856,297 | 35,918,334 |
| qTA4-1 | 4 | GLM-whole | 7,23,871 | 6,54,721 | 7,24,006 |
| qTA4-2 | 4 | MLM/GLM-whole | 9,72,323 | 9,21,112 | 9,72,402 |
| qTA4-3 | 4 | GLM-whole | 1,902,732 | 1,886,204 | 1,979,550 |
| qTA5-1 | 5 | MLM-ind | 17,795,916 | 17,795,916 | 18,061,792 |
| qTA5-2 | 5 | MLM-ind | 18,668,045 | 18,485,378 | 18,709,357 |
| qTA5-3 | 5 | MLM-ind | 19,108,377 | 19,108,377 | 19,210,385 |
| qTA6 | 6 | MLM-ind | 26,092,692 | 26,086,626 | 26,127,844 |
| qTA8-1 | 8 | MLM-ind | 20,690,040 | 20,689,044 | 20,696,741 |
| qTA8-2 | 8 | GLM-whole | 20,802,316 | 20,802,316 | 20,995,468 |
| qTA8-3 | 8 | MLM-ind | 28,327,637 | 28,326,448 | 28,438,469 |
| qTA9-1 | 9 | GLM-jap | 1,193,941 | 1,070,773 | 1,193,941 |
| qTA9-2 | 9 | MLM/GLM-Jap | 2,070,634 | 1,313,684 | 2,207,902 |
| qTA9-3 | 9 | MLM/GLM-Jap | 2,516,331 | 2,407,587 | 2,621,693 |
| qTA9-4 | 9 | MLM/GLM-Jap | 2,980,680 | 2,936,852 | 3,180,164 |
| qTA9-5 | 9 | GLM-jap | 3,383,978 | 3,383,859 | 3,486,825 |
| qTA9-6 | 9 | *MLM-whole | 20,739,050 | 20,721,611 | 20,749,797 |
| qTA10-1 | 10 | *MLM/GLM-whole | 14,414,010 | 14,191,834 | 14,733,631 |
| qTA10-2 | 10 | GLM-jap | 20,273,073 | 20,268,702 | 20,295,300 |
| qTA10-3 | 10 | MLM/GLM-Jap | 22,799,648 | 22,774,612 | 22,836,250 |
| qTA11-1 | 11 | GLM-whole | 4,826,905 | 4,826,905 | 4,829,616 |
| qTA11-2 | 11 | GLM-ind | 13,090,765 | 13,090,765 | 13,171,938 |
| qTA12-1 | 12 | MLM-ind | 3,329,842 | 3,258,584 | 3,539,326 |
| qTA12-2 | 12 | MLM-ind | 9,262,194 | 9,262,194 | 9,337,250 |
| qTA12-3 | 12 | MLM-ind | 9,606,800 | 9,606,800 | 9,755,631 |
| qTA12-4 | 12 | MLM/GLM-whole | 21,905,686 | 21,901,194 | 22,047,563 |

**Note:**
* These QTLs were also identified and confirmed by Non-Syn GWAS.

ten functional genes on the basis of protein change and expression variation were observed for tiller angle (Fig. 5, Fig. S1). The *Os10g0412700* showed the maximum range of tiller angle (37.50–48.14) among haplotypes while *Os01g0621600* have maximum haplotypes variation on the basis of three SNPs (Fig. 5).

For the known gene *TAC1*, three SNPs in CDS region were observed as the functional to develop four and three haplotypes in *indica* and *japonica* sub-populations respectively (Fig. 6, Table S1). The *TAC1* has already known to cause the functional variation for tiller angle. The identification of its superior haplotype would be beneficial for selection of compact plants and worth further studies.

## DISCUSSION

The plant morphological traits have important contribution to the total yield and resistance against lodging. The elite architecture may include short stature, thick and strong culm, fewer unproductive tillers, more number of panicles with more grains per panicle and erect type of leaves (*Khush, 1995*). Most of these traits including tiller angle in rice are proposed to govern by quantitative trait loci. Different tiller angle governing QTLs have been identified by studies based on different mapping material. In this study, we performed the GWAS analysis using a huge natural population. A wide range of tiller angle of cultivated species was observed in diverse population. Furthermore, the 13X deep sequence data provided a high resolution for fine-scale mapping in GWAS. The deep sequence data assembled with a sufficient number of land races also acquired the advantage of maximum genetic diversity which was targeted to explore in GWAS. In this study, an effort was made to improve the mapping resolution, imputation efficiency and identification of new alleles by addition of new SNPs in the worldwide collection of accessions for genome wide association study (*Huang et al., 2010*).

No doubt GWAS is an effective tool to explore the complex genetic variation of rice genome but it is still pretentious because of resulting in identification of false positives (*Atwell et al., 2010*). The computation of structural associations and genetic control are two approaches to avoid the identification of false positive signals in human and plants (*Devlin & Roeder, 1999*; *Marchini et al., 2004*; *Yu et al., 2006*). In this study, the whole association panel was divided into two subpopulations and re-evaluated separately as *indica* and *japonica* to control the population structure based false positives. In addition, the population structure and genetic relatedness among individuals were uncovered by PCA and kinship matrix in GAPIT program that may helpful to control the type I error (false positives). Finally, we used the GLM method with Q-matrix (PCA) and compressed mix linear model (cMLM) with Q and K (kinship) matrixes for three structural panels; whole population, *indica* and *japonica* subpopulations, as it has been suggested to reduce the false positives (*Yang et al., 2010*; *Yang et al., 2011*). Along with structural differentiation into three panels and adoption of two different models to avoid false positives, 1,000 permutation test was also performed with whole population in cMLM and a significance threshold of $-\log10 (P) \geq 4$ was set to reveal the true associations (Fig. 3).

| Sub population | haplotype | Os01g0621600 CDS 24786734 | Os01g0621600 CDS 24786776 | Os01g0621600 Promoter 24789507 | N | TA |
|---|---|---|---|---|---|---|
| Indica | hap1 | A | C | A | 175 | 44.08 |
| Indica | hap2 | T | C | A | 14 | 38.93 |
| Indica | hap3 | T | T | A | 132 | 39.42 |
| Indica | hap4 | T | T | G | 34 | 38.89 |
| | | | | | | |
| Japonica | hap1 | A | C | A | 24 | 39.82 |
| Japonica | hap2 | T | C | A | 10 | 38.23 |
| Japonica | hap3 | T | T | A | 25 | 37.59 |
| Japonica | hap4 | T | T | G | 148 | 37.21 |

| Sub population | haplotype | Os10g0412700 CDS 14399366 | Os10g0412700 promoter 14402688 | N | TA |
|---|---|---|---|---|---|
| Indica | hap1 | C | A | 39 | 48.14 |
| Indica | hap2 | C | G | 320 | 42.13 |
| Indica | hap3 | T | G | 58 | 38.01 |
| | | | | | |
| Japonica | hap2 | C | G | 64 | 39.87 |
| Japonica | hap3 | T | G | 156 | 37.50 |

**Figure 5 Haplotypes of two significant loci for tiller angle in *indica* and *japonica* subpopulations.**

LD determination is very important to examine the resolution of association analysis (*Flint-Garcia, Thornsberry & Buckler, 2003*). Different studies have been performed to evaluate the LD decay in rice, and 167 kb and 123 kb LD decay rate in *japonica* and *indica* subpopulations have already been reported previously (*Huang et al., 2010*; *Huang et al., 2012*). Therefore, to maintain the power of GWAS resolution in present study, the criteria of minimum three SNPs in 170 kb LD range was strictly followed for QTL declaration.

To date, some tiller angle related QTLs/genes have been reported by linkage mapping. Contrastingly, association mapping not only provided a higher mapping resolution but also helped to detect the candidate loci (*Li et al., 2003*; *Yu et al., 2007*; *Jin et al., 2008*). Therefore, GWAS has widely been performed to disclose the genetic variation of morphological traits in rice (*Huang et al., 2010*; *Kumar et al., 2015*). Similarly in our study, a total 336 significant SNPs classified into 37 QTLs which according to Rice Genome Annotation Project database include 93 annotated loci were identified for tiller angle (Table S2). Among these QTLs a known gene *TAC1* was observed in *qTA9-6* which can validate the performance of GWAS. Other known tiller angle related genes were not observed in association signals that may due to over missing or insufficient SNPs availability in specific genomic region. Another possible reason was type II error (false negatives) due to strict threshold.

We introduce a sequence pooling strategy to identify the significant loci. The major and minor allele frequencies were compared between the genotypes in high and low phenotype pools. The 28 SNPs in 12 QTLs which included the 22 annotated loci were screened in this way. To identify the functional variations in these loci, all the SNPs included in CDS and promoter were used and functional haplotypes were evaluated. The single significant gene per QTL and its functional variation will be validated by linkage population and genetic transformation in future studies.

As the significant difference in tiller angle between *japonica* and *indica* genotypes has been observed and reported, this study demonstrated the feasibility to perform the genome wide association analysis for a morphological trait which is vulnerable to population structure. Because of substantial resolution power, it also revealed that the dissection of

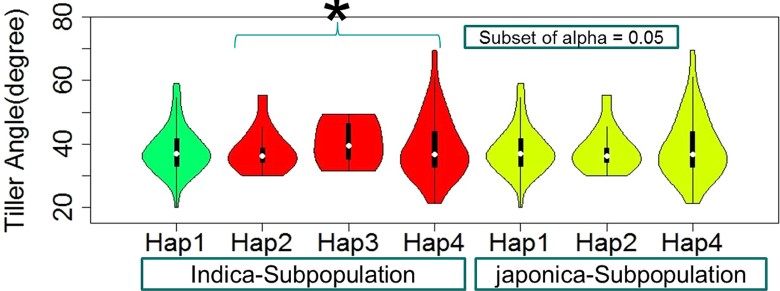

| sub-population | Haplotype | 20733643 | 20733730 | 20734345 | No. of cultivars | Tiller Angle(°) |
|---|---|---|---|---|---|---|
| Indica | Hap-1 | A | T | A | 133 | 38.94[a] |
| | Hap-2 | A | C | A | 22 | 39.60[b] |
| | Hap-3 | A | C | G | 5 | 40.44[b] |
| | Hap-4 | T | C | G | 344 | 43.04[b] |
| | | | | | | |
| Japonica | Hap-1 | A | T | A | 208 | 37.82[a] |
| | Hap-2 | A | C | A | 13 | 37.91[a] |
| | Hap-4 | T | C | G | 52 | 39.05[a] |

**Figure 6 Haplotypes of known gene *TAC1* significant loci for tiller angle in *indica* and *japonica* subpopulations, hence an asterisk (\*) indicates the haplotypes with wider tiller angle.**

complex quantitative traits can be initiated from QTL mapping that can further confirmed by linkage mapping and genetic transformation of specific genomic region. The genome wide association study for tiller angle in rice populations provided the significant loci. The manipulation of these loci can improve the plant architecture and can be used for ideotype breeding. These results provided the striking information for development of further breeding programs for lodging resistance and higher yield.

## CONCLUSIONS

The improvements in rice plant architecture especially the tiller angle allows the efficient light capture by increasing planting density. The deep insight into genetic basis of plant architecture will help to plan the future breeding programs. Current study revealed 37 significant QTLs with 93 annotated loci governing the tiller angle in rice. The accuracy of results was proved by the identification of a known tiller angle controlling locus *TAC1*. The sequence pooling technique was observed helpful to screen the 12 significant QTLs with 22 annotated loci. The functional variations were estimated by haplotype analysis and the genotypes were screened for significant haplotypes. These results deepened our understanding about genetic basis of tiller angle in rice. The findings can play a significant role in future breeding programs.

## ACKNOWLEDGEMENTS

We thank the supports from the union laboratory between Guangxi key laboratory of rice genetics and breeding and state key laboratory of agrobiotechnology.

### Funding

This work was supported by grants from the Ministry of Science and Technology of China (2021YFD1200502), the national natural science foundation of China (31801324, 32072036, 32001521), and the project of the central government guides the development of local science and technology (GuikeZY20198015). The funders had no role in study design, data collection and analysis, decision to publish, or preparation of the manuscript.

### Grant Disclosures

The following grant information was disclosed by the authors:
Ministry of Science and Technology of China: 2021YFD1200502.
National Natural Science Foundation of China: 31801324, 32072036, 32001521.
Development of Local Science and Technology: GuikeZY20198015.

### Competing Interests

Rana Muhammad Atif is an Academic Editor for PeerJ. The authors declare that they have no competing interests.

### Author Contributions

- Muhammad Abdul Rehman Rashid conceived and designed the experiments, performed the experiments, analyzed the data, prepared figures and/or tables, authored or reviewed drafts of the paper, and approved the final draft.
- Rana Muhammad Atif conceived and designed the experiments, analyzed the data, prepared figures and/or tables, authored or reviewed drafts of the paper, and approved the final draft.
- Yan Zhao performed the experiments, analyzed the data, prepared figures and/or tables, authored or reviewed drafts of the paper, and approved the final draft.
- Farrukh Azeem performed the experiments, analyzed the data, prepared figures and/or tables, authored or reviewed drafts of the paper, and approved the final draft.
- Hafiz Ghulam Muhu-Din Ahmed conceived and designed the experiments, analyzed the data, prepared figures and/or tables, authored or reviewed drafts of the paper, and approved the final draft.
- Yinghua Pan conceived and designed the experiments, analyzed the data, prepared figures and/or tables, authored or reviewed drafts of the paper, and approved the final draft.
- Danting Li performed the experiments, analyzed the data, authored or reviewed drafts of the paper, and approved the final draft.
- Yong Zhao performed the experiments, analyzed the data, authored or reviewed drafts of the paper, and approved the final draft.
- Zhanying Zhang analyzed the data, authored or reviewed drafts of the paper, and approved the final draft.

- Hongliang Zhang conceived and designed the experiments, authored or reviewed drafts of the paper, and approved the final draft.
- Jinjie Li conceived and designed the experiments, authored or reviewed drafts of the paper, and approved the final draft.
- Zichao Li conceived and designed the experiments, authored or reviewed drafts of the paper, and approved the final draft.

## Data Availability

Raw sequence data is available at Sequence Read Archive (SRA): PRJEB6180.

## Supplemental Information

Supplemental information for this article can be found online at http://dx.doi.org/10.7717/peerj.12674#supplemental-information.

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
