# Peer review of "Dissection of genetic architecture for tiller angle in rice (*Oryza sativa*. L) by multiple genome-wide association analyses"

_PeerJ, doi:10.7717/peerj.12674_

## Round 0.1 · original submission · Major Revisions

The paper is in good shape and describes very interesting results. However, going through the reviewer's comments and paper itself, I feel that it needs significant improvement. Please answer point-to-point queries of the reviewers. Also try to address the following points:

1. Recent literature is missing in the “Introduction” and “Discussion” sections. Authors are needed to provide some recent references, especially from the last five years.

2. The tiller angle is a very important trait in rice and is the major component of the paper. Unfortunately, no impressive reference related to tiller angle is available in the paper

3. Language needs to be improved. Authors are required to thoroughly read the paper and look into grammatical and spelling mistakes. Moreover, there are some overlapping sentences in the abstract and conclusion. Please make them correct

4. Discussion needs to be improved with reference to Os10g0412700 and Os01g0621600 genes

Reviewer 1 ·

Basic reporting

The authors provided an exhaustive list of references in the place. but lack of recent literature. For example: TIG1 gene,Weifeng Zhang, Lubin Tan (2019), Molecular Plant, 12(8): 1075-1089.

Experimental design

no comment

Validity of the findings

no comment

Additional comments

1、 Your introduction needs more detail. I suggest that you improve the description at lines 26-28 to provide more justification for your study (specifically, you should expand upon the knowledge gap being filled).
Tiller angle is one of the most important factors affecting rice plant architecture.
 Plant architecture, a collection of the important agronomic traits that determine grain production in rice, is mainly affected by factors including tillering, plant height, leaf shape and arrangement, and panicle morphology. (Yonghong Wang, Jiayang Li. (2005) Plant Mol Biol. Sep;59(1):75-84.)
 Plant architecture, usually referred to as the three-dimensional organization of the aerial part of a plant, is mainly determined by factors including branching (tillering) pattern, plant height, leaf shape and arrangement, and inflorescence morphology (Reinhardt, D. and Kuhlemeier, C. (2002) Plant architecture. EMBO Rep. 3, 846–851. Wang, Y. and Li, J. (2006) Genes controlling plant architecture. Curr. Opin. Biotechnol. 17, 123–129.
2、 lines 60-73: The authors provided an exhaustive list of references in the place. but lack of recent literature. For example: TIG1 gene,Weifeng Zhang, Lubin Tan (2019), Molecular Plant, 12(8): 1075-1089.
3、 There are many errors in the manuscript, such as, Lines 146-147: “positively semi-skewed frequency”, Figure 1. No normal distribution curves, Table 1. “skewness”. Lines 170-171: “Among these QTLs, 13 and 10 QTLs were specifically identified in indica and japonica subpopulations while 14 non-common QTLs were observed in whole population.” Table 3, japonica:9, 15 non-common QTLs. Please check the manuscript carefully.

Reviewer 2 ·

Basic reporting

Rice is one of the most important crops in the world. Tiller angle is an important factor affecting yield. In this work, Muhammad Abdul Rehman Rashid et al. performed genome wide association analysis by GLM and cMLM for three populations. Thirty-seven QTLs with 93 annotated loci were identified. Among the loci, a known tiller angle controlling locus TAC1 was also identified. The introduction of sequence pooling technique was observed fruitful to screen the 12 significant QTLs with 22 annotated loci. The research method is reasonable and good research results are obtained.

Experimental design

no comment

Validity of the findings

no comment

Additional comments

Q1: The paper should analysis protein domain of candidate gene Os10g0412700 and Os01g0621600. And the genes function should be analysis.
Q2. In discussion, the candidate genes Os10g0412700 and Os01g0621600 should be discussed with relevant literature.
Q3. The language and writing of the paper still need to be polished. For example, there are too many similar sentences between Abstract and Conclusions.

Reviewer 3 ·

Basic reporting

The research is on QTLs and genes mapping for tiller angle in rice. The manuscript is clear and well-drafted to understand.
The overall structure of the manuscript and raw material provided is unambiguous. The defined problem and hypothesis of the research are well addressed in the results and discussion.
However, some most recent studies on tiller angle genes and QTLs are not cited in the literature review. I suggest to cite the important genes/QTLs including, LAZY2, qTAC-8, TAC-3, TAC-4, TIG1, OsHOX28 etc in this regards

Experimental design

the research belongs to the tiller angle in rice as a model plant that is directly related to the scope of the journal. The research purpose and problem are clearly defined. the methodology is well explained and the statistical models the clearly explained.

Validity of the findings

The research is replicated multiple times and findings are reproducible. the genetic and phenotypic information is clearly provided.
the conclusion part is well stated.

Additional comments

Additionally, it's observable that there was no recent research of this tiller angle has been cited.
It's highly recommended to add some newer investigations to identify the gap between previous and present studies.

---

## Round 0.2 · Minor Revisions

Gerard Lazo, the Section Editor, has commented and said:

"The manuscript may be OK, but there is a need to refine the clarity in presentation. It’s not too bad, but I am unwilling to do the proofing when it should be done by someone more familiar with the project. I have included some markup (attached). Once refined I will re-assess, but the general impression seems to indicate that there may be a ready acceptance. I recommend further revision."

Please make the revisions as suggested by Section Editor.

Reviewer 1 ·

Basic reporting

no comment

Experimental design

no comment

Validity of the findings

no comment

Reviewer 2 ·

Basic reporting

In the first round of review, the authors have made significant changes. Now the draft is much clear and acceptable to publish.

Experimental design

Good enough

Validity of the findings

Experiments are well designed and results are clear and repeatable. Shortly, results are valide.

Additional comments

the authors have made the linguist and technical corrections as suggested.

Reviewer 3 ·

Basic reporting

The version2 is satisfactor and authors have addressed all my comments.

Experimental design

Well written in v2

Validity of the findings

Valid and statistically robust

---

## Round 0.3 · accepted · Accept

The paper is much improved now and is acceptable